# The observational EURACAN prospective clinical registry dedicated to epithelioid hemangioendothelioma: The protocol of an international and collaborative effort on an ultra-rare entity

Anna Maria Frezza[1], Hugh Leonard[2], Ninna Aggerholm-Pedersen[3], Giuseppe Badalamenti[4], Paolo Baili[5], Giacomo G. Baldi[6], Sebastian Bauer[7], Serena Bazzurri[6], Irene Benzonelli[8], Alexia Bertuzzi[9], Jean-Yves Blay[10], Giuseppe Bianchi[11], Simone Bonfarnuzzo[5], Christophe Bouvier[10], Kyetil Boye[12], Javier Martin Broto[13], Antonella Brunello[14], Domenico Campanacci[15], Paolo G. Casali[1], Carlo Cicala[16], Elisa Crotti[9], Lorenzo D'Ambrosio[8], Angelo Paolo Dei Tos[17], Nils Dieckmann[7], Armelle Dufresne[10], Stephanie Elston[18], Virginia Ferraresi[19], Stefano Gabellini[6], Claudia Giani[1], Vincenzo Giannusa[4], Melissa Gil Sanjines[20], Teresa Grassani[21], Alessandro Gronchi[22], Paolo Lasalvia[20], Stefan Lindskog[23], Nadia Hindi[13], Matilde Ingrosso[1], Andrei Ivanescu[24], Robin Jones[18], Iwona Lugowska[25], Julia Ketzer[7], Anna Mariuk-Jarema[25], Alessandro Mazzocca[21], Laura Monteleone[1], Carlo Morosi[26], Andrea Napolitano[18], Francesca Nardozza[27], Elisabetta Neri[15], Maria Nilsson[23], Andri Papakonstantinou[28,29], Sandro Pasquali[30], Marta Sbaraglia[17], Federico Scolari[15], Joanna Szkandera[31], Claudia Valverde[16], Bruno Vincenzi[21], Salvatore Vizzaccaro[14], Federica Zuccheri[11], Silvia Stacchiotti[1], Annalisa Trama[20]*

1 Fondazione IRCCS Istituto Nazionale Tumori, Medical Oncology 2, Milan, Italy, 2 EHE Rare Cancer Charity, London, United Kingdom, 3 Department of Oncology, Aarhus University Hospital, Aarhus, Denmark, 4 Department of Surgical, Oncological and Oral Sciences, Section of Medical Oncology, University of Palermo, Palermo, Italy, 5 Department of Epidemiology and Data Science, Fondazione IRCCS Istituto Nazionale Tumori, Data Science Unit, Milan, Italy, 6 Department of Oncology, Hospital of Prato, Azienda USL Toscana Centro, Prato, Italy, 7 Medical Oncology, University Hospital Essen, Essen, Germany, 8 Medical Oncology, Ospedale San Luigi, Orbassano, Italy, 9 Medical Oncology, IRCCS, Humanitas Research Hospital, Rozzano-Milano, Italy, 10 Medical Oncology, Centre Léon Bérard, Lyon, France, 11 IRCCS Istituto Ortopedico Rizzoli, Clinica Ortopedica e Traumatologica III a Prevalente Indirizzo Oncologico, Bologna, Italy, 12 Medical Oncology, Oslo University Hospital, Oslo, Norway, 13 Medical Oncology, Fundación Jimenez Diaz University Hospital, Madrid, Spain, 14 Istituto Oncologico Veneto IOV—IRCCS, Medical Oncology 1, Padua, Italy, 15 Orthopaedic Oncology, Careggi University Hospital, Florence, Italy, 16 Medical Oncology, Hospital Universitari Vall d'Hebron, Spain, 17 Department of Pathology, Azienda Ospedaliera di Padova, Padua, Italy, 18 Sarcoma Unit, The Royal Marsden NHS Foundation Trust, London, United Kingdom, 19 Sarcomas and Rare Tumors Departmental Unit-IRCCS Regina Elena National Cancer Institute, Rome, Italy, 20 Department of Epidemiology and Data Science, Fondazione IRCCS Istituto Nazionale Tumori, Evaluative Epidemiology Unit, Milan, Italy, 21 Medical Oncology, Università Campus Bio-Medico, Rome, Italy, 22 Fondazione IRCCS Istituto Nazionale Tumori, Sarcoma Surgery, Milan, Italy, 23 Department of Surgery, Institute of Clinical Sciences, Sahlgrenska Academy, University of Gothenburg, Gothenburg, Sweden, 24 EHE Italia, Milan, Italy, 25 Maria Skłodowska Curie Institute—Oncology Centre, Early Phase Trial Unit, Warsaw, Poland, 26 Department of Radiology, Fondazione IRCCS Istituto Nazionale dei Tumori, Milan, Italy, 27 IRCCS Regina Elena National Cancer Institute, UOSD Clinical Trial Center, Biostatistics and Bioinformatics, Rome, Italy, 28 Department of Oncology-Pathology, Karolinska Institutet, Stockholm, Sweden, 29 Department of Breast Cancer, Endocrine Tumors and Sarcoma, Karolinska Comprehensive Cancer Center, Karolinska University Hospital, Stockholm, Sweden, 30 Department of Experimental Oncology, Fondazione IRCCS Istituto Nazionale dei Tumori, Molecular Pharmacology Unit, Milan, Italy, 31 Department of Internal Medicine, Division of Oncology, Medical University of Graz, Graz, Austria

* annalisa.trama@istitutotumori.mi.it



**Data Availability Statement:** Access to the EURACAN registry on EHE is based on governance

set with EURACAN member and are reported below. Data Access Rules: 1) data remain the property of the contributing health care providers (HCP) 2) each HCP is free to access and use its own data for research purposes 3) each HCP including data in the EURACAN Registry can request access to the EURACAN Registry upon the presentation of a study protocol that has to be approved by the Steering Committee of the Registry 4) third parties (e.g., pharmaceutical companies, patient organisations, competent authorities etc.) can request use of the registry data. However, data should be used for projects that include EURACAN members. Thus, third parties proposing a research question, should work with a EURACAN Principal Investigator (PI) and should present a study protocol to be reviewed by the SC. If the research question is relevant but does not require a study protocol, a quick review will be provided by the SC and a written report including results of data analyses will be shared with the third party 5) Commercial companies, depending on the study, may be asked to contribute funding. Funding will be used to support the registry maintenance and/or specific studies proposed by the EURACAN members and based on the registry data 6) each EURACAN domain will define a domain registry working group (RWG), made up of 3 to 5 members including a patient representative, to review the study protocols presented for the domain. Additional expertise may be brought in as required. The domain leader will chair this WG and will report to the SC the results of the revision to inform and get SC approval 7) once a study on a specific domain is approved by the SC, the scientific secretariat will inform the relevant HCP by email requesting to express its willingness to share its data for the approved study. If an HCP does not wish to share its data for the approved study, it can opt out from the study. In case, it is expected that it will provide explicit reason. The HCP should inform the scientific secretariat within 2 weeks. In the event the HCP does not confirm whether it wishes to share its data within 2 weeks from the scientific secretariat request, its data will not be used for the study with the vision of EURACAN being a network of networks and therefore also a registry network 8) the scientific secretariat will inform the principal investigator (PI) of the proposed study about the SC decision 9) the PI of the proposed study will be responsible for arranging a preliminary ethical review that will be shared with the participating centres. The participating centres should provide their institutional ethical review response to the PI within 60 days 10) the engagement of the EURACAN Registry in international collaborative efforts is highly supported. In this context, the SC

# Abstract

## Introduction

Epithelioid hemangioendothelioma (EHE) is an ultra-rare sarcoma, marked by distinctive molecular and pathological features and with a variable clinical behavior. Its natural history is still partially understood, reliable prognostic and predictive factors are lacking and many questions are still open on the optimal management. In the context of EURACAN, a prospective registry specifically dedicated to EHE was developed and launched with the aim of providing, through high-quality prospective data collection, a better understanding of this disease.

## Study design

Registry-based cohort study including only new cases of patients with a pathological and molecularly confirmed diagnosis of EHE

## Objectives

To improve the understanding of EHE natural history, validate and identify new prognostic and predictive factors, clarify the activity and efficacy of currently available treatment options, describe treatment pattern.

## Methods

### Settings and participants

It is an hospital-based registry established in centers with expertise in EHE including adult patients with a new pathological and molecularly confirmed diagnosis of EHE starting from the 1st December 2023. The characteristics of each patient in the facility who meets the above-mentioned inclusion criteria will be collected prospectively and longitudinally with follow-up at cancer progression and / or cancer relapse or patient death. It is a secondary use of data which will be collected from the clinical records. The data collected for the registry will not entail further examinations or admissions to the facility and/or additional appointments to those normally provided for routine patient follow-up.

### Variables

Full details on patients and disease features, treatment and outcome will be collected, according to common clinical practice guidelines developed and shared with all the contributing centers. In addition, data on potential confounders (e.g. comorbidity; functional status etc.) will also be collected.

### Statistical methods

The data analyses will include descriptive statistics and analytical analyses. Multivariable Cox's proportional hazards model and Hazard ratios (HR) for all-cause or cause-specific mortality will be used to determine independent predictors of overall survival, recurrence and progression.

is asked to deliberate on collaborative projects involving the registry. Also, in this case HCPs and national registries will be informed and will be asked to agree or to opt-out of such engagement.

**Funding:** The author(s) received no specific funding for this work.

**Competing interests:** Anna Maria Frezza declares institutional research funding from Advenchen Laboratories, Amgen Dompé, AROG Pharmaceuticals, Ayala Pharmaceuticals, Bayer, Blueprint Medicines, Boehriger Ingelheim, Daiichi Sankyo, Deciphera, Eisai, Eli Lilly, Epizyme Inc, Foghorn Therapeutics Inc., Glaxo, Hutchison MediPharma Limited, Inhibrx, Inc., Karyopharm Pharmaceuticals, Novartis, Pfizer, PharmaMar, PTC Therapeutics, Rain Oncology, SpringWorks Therapeutics. Alexia Bertuzzi declares support for attending meeting and/or travels from Pharmamar and Istituto Gentili; support for scientific activities from Istituto Gentili. Antonella Brunello declares consulting fees or serving on advisory boards for Eli Lilly, Roche, GSK, Eisai, Pharmamar, Boehringer Ingelheim, Deciphera; payment or honoraria for educational events by GSK and Pharmamar; travel grants by Pharmamar, Istituto Gentili. Giacomo G. Baldi declares consulting fees from Eli Lilly, Pharmamar, AboutEvents; honoraria from Pharmamar, Eli Lilly, Glaxo Smith Kline, Merck Sharp & Dome, Eisai, IstitutoGentili; support for attending meetings and/or travels from Novartis, Pharmamar, Eli Lilly; participation on the advisory board from Pharmamar, Eli Lilly, Glaxo Smith Kline, Merck Sharp & Dome, Eisai. Paolo G. Casali declares institutional research funding from Advenchen Laboratories, Amgen Dompé, AROG Pharmaceuticals, Ayala Pharmaceuticals, Bayer, Blueprint Medicines, Boehriger Ingelheim, Daiichi Sankyo, Deciphera, Eisai, Eli Lilly, Epizyme Inc, Foghorn Therapeutics Inc., Glaxo, Hutchison MediPharma Limited, Inhibrx, Inc., Karyopharm Pharmaceuticals, Novartis, Pfizer, PharmaMar, PTC Therapeutics, Rain Oncology, SpringWorks Therapeutics. Virginia Ferraresi declares consultancy fees or honoraria from PharmaMar, Boehringer Ingelheim, Gentili e Serb Pharmaceuticals. Matilde Ingrosso reports institutional research funding from Advenchen Laboratories, Amgen Dompé, AROG Pharmaceuticals, Ayala Pharmaceuticals, Bayer, Blueprint Medicines, Boehriger Ingelheim, Daiichi Sankyo, Deciphera, Eisai, Eli Lilly, Epizyme Inc, Foghorn Therapeutics Inc., Glaxo, Hutchison MediPharma Limited, Inhibrx, Inc., Karyopharm Pharmaceuticals, Novartis, Pfizer, PharmaMar, PTC Therapeutics, Rain Oncology, SpringWorks Therapeutics. Laura Monteleone reports

## Results

The registry has been joined by 21 sarcoma reference centers across EU and UK, covering 10 countries. Patients' recruitment started in December 2023. The estimated completion date is December 2033 upon agreement on the achievement of all the registry objectives. The already established collaboration and participation of EHE patient's associations involved in the project will help in promoting the registry and fostering accrual.

## Introduction

Given its incidence of 0.038/100 000/year, EHE is an ultra-rare sarcoma, with distinctive, well-defined pathological, molecular and clinical features [1,2]. It belongs to the group of vascular sarcomas and is characterized by *WWTR1-CAMTA1* (90%) or *YAP1*-TFE3 (10%) gene fusions, which represents today a hallmark for diagnosis. EHE potentially arises everywhere in the body and shows a high tendency toward metastatic spread, especially in the lung, liver and bone [3,4]. The onset is characterized by different presentations, including a unifocal lesion (usually in the soft tissues), locoregional metastases (multiple lesions in a single organ or in a single anatomic compartment) and systemic metastases (multi-organ involvement) [4]. Also, the biological behavior of the disease is unique in sarcomas and variable, with spontaneous regressions reported, patients with untreated stable disease overtime, slowly progressive variants and highly aggressive and rapidly fatal cases. Today, the relative incidence of the different presentations is undefined, the natural history of the different subtypes is poorly understood, and reliable clinical or biological prognostic factors are lacking. Retrospective data suggest that patients with EHE presenting or developing during their course serosal involvement and / or effusion, systemic associated symptoms and anemia have a worse prognosis, but a prospective validation of this observation is lacking [5–7]. In terms of management, surgery is the treatment of choice for localized disease, with an excellent outcome (expected cure rate of 70–80%) and no proven role for local or systemic adjuvant therapies. For patients with advanced, asymptomatic disease, active surveillance is often the upfront choice in order to minimize the risk of overtreatment [4]. Systemic therapies are usually considered for patients with progressive or symptomatic disease, although a standard medical approach is currently not established. Unfortunately, conventional chemotherapy, including anthracycline-based regimens widely regarded as the mainstay in the treatment of advanced soft tissue sarcomas, showed marginal activity in EHE [8]. The use of potentially active compounds, such as *m-TOR* inhibitors which proved the highest activity in EHE and could therefore represent the preferred option, is limited by regulatory constraints as they are not currently labeled in Europe for this indication [4,6,9]. Given the degree of uncertainty on EHE management, a global consensus meeting was organized in December 2020 under the umbrella of the European Society for Medical Oncology (ESMO) involving > 80 multidisciplinary, worldwide experts together with a patient representative, with the aim of defining, by consensus, evidence-based best practices for the optimal approach to primary and metastatic EHE [4].

Since the number of patients is inherently low at the hospital level, the only way to increase knowledge about EHE and improve the diagnosis, treatment, and prognosis of this complex and heterogeneous disease is to harmonize and combine real-world data from sarcoma expert centers by leveraging a wide, international, joined effort. In this context, a unique opportunity is provided by the European Reference Networks (ERNs), virtual network established by the EU commission aiming to tackle rare conditions, including cancers. Among the three ERNs

institutional research funding from Advenchen Laboratories, Amgen Dompé, AROG Pharmaceuticals, Ayala Pharmaceuticals, Bayer, Blueprint Medicines, Boehriger Ingelheim, Daiichi Sankyo, Deciphera, Eisai, Eli Lilly, Epizyme Inc, Foghorn Therapeutics Inc., Glaxo, Hutchison MediPharma Limited, Inhibrx, Inc., Karyopharm Pharmaceuticals, Novartis, Pfizer, PharmaMar, PTC Therapeutics, Rain Oncology, SpringWorks Therapeutics. Johanna Szkandera reports participation in advisory boards or invited speaker fees for PharmaMar, Bayer, Roche, Lilly, Amgen; travel expenses coverage from PharmaMar, Roche, Lilly, Amgen, Bristol Myers Squibb; research funding from PharmaMar, Roche, Eisai. Bruno Vincenzi reports consulting fees from Eisai, Lilly, Bayer, Deciphera, PharmaMar, Blueprint, Pfizer, GSK, Accord, Abbott and research support from PharmaMar, Novartis, Lilly Silvia Stacchiotti declares advisory board roles with Bayer, Boehringer, Daiichi, Ikena, Nec Oncology, Pharma Essentia, Regeneron, Servier; invited speaker roles for Bayer, Boehringer, Gentili, Pharmamar; institutional research funding from Advenchen Laboratories, Amgen Dompé, AROG Pharmaceuticals, Ayala Pharmaceuticals, Bayer, Blueprint Medicines, Boehriger Ingelheim, Daiichi Sankyo, Deciphera, Eisai, Eli Lilly, Epizyme Inc, Foghorn Therapeutics Inc., Glaxo, Hutchison MediPharma Limited, Inhibrx, Inc., Karyopharm Pharmaceuticals, Novartis, Pfizer, PharmaMar, PTC Therapeutics, Rain Oncology, SpringWorks Therapeutics. Carlo Cicala, Lorenzo D'Ambrosio, Angelo Paolo Dei Tos, Alessandro Gronchi, Nadia Hindi, Robin Jones, Iwona Lugowska, Julia Ketzer, Anna Mariuk-Jarema, Andrea Napolitano, Andri Papakonstantinou, Sandro Pasquali, Marta Sbaraglia, Federico Scolari, Elisa Crotti, Irene Benzonelli, Jean-Yves Blay, Christophe Bouvier, Javier Martin Broto, Hugh Leonard, Ninna Aggerholm-Pedersen, Andri Papakonstantinou, Giuseppe Bianchi, Federica Zuccheri, Paolo Baili, Simone Bonfarnuzzo, Domenico Campanacci, Armelle Dufresne, Sebastian Bauer, Serena Bazzurri, Stefano Gabellini, Claudia Giani, Vincenzo Giannusa, Melissa Gil-Sanjines, Teresa Grassani, Paolo Lasalvia, Stefan Lindskog, Salvatore Vizzaccaro, Nils Dieckmann, Alessandro Mazzocca, Andrei Ivanescu, Giuseppe Badalamenti, Carlo Morosi, Stephanie Elston, Kyetil Boye, Francesca Nardozza, Elisabetta Neri, Maria Nilsson and Annalisa Trama declares no conflict of interest.

dedicated to cancer, EURACAN (https://euracan.eu) is the one focusing on the 10 families of rare adult solid cancers (RACs), including soft tissue sarcomas, bone sarcomas and gastrointestinal stromal tumour. Fostering academic research and promote epidemiological surveillance in rare cancers through setting up of shared registries is recognize as one of ERNs priorities. This is why within EURACAN the EU-supported project named "Starting an Adult Rare Tumour Registry (STARTER)" began on April 1st 2020 to develop the EURACAN registry dedicated to the RACs (https://euracan.eu/registries/starter/). The registry it is now running, and since May 2022 is open to data collection on rare head and neck cancers (ClinicalTrials.gov Identifier: NCT05483374). Within the sarcoma domain, taking into account the challenges described above, priority was given to ultra-rare sarcoma, and thanks to the strong connection, motivation and support provided by EHE Rare Cancer Charity UK, it was decided to start from EHE. We believe that in this context the collection of high-quality data by clinical registries will be crucial to improve the understanding of EHE natural history, validate and identify new prognostic factors, clarify the activity and efficacy of currently available treatment options and eventually provide and external control which might serve as a benchmark for the development of new drugs and support discussion with regulatory authorities.

EHE registry objectives:

- To describe EHE population and diseases characteristics at diagnosis

- To define EHE natural history, as a whole and in its variants

- To describe EHE response to different systemic therapies and radiation therapy

- To assess effectiveness of different treatments (surgery, radiation therapy, systemic therapies, ILP, locoregional techniques and combination)

- Identify prognostic and predictive factors including:

○ Age at diagnosis, gender, hormonal status (for females), performance status

○ Disease extent at presentation, disease size (for primary localized tumours) and disease site [4,7,10,11]

○ Signs and symptoms at presentation and during follow up, including tumour-related pain, systemic symptoms (such as temperature, asthenia, anorexia, weight loss, night sweating) and limb edema [7]

○ Anaemia and fibrinogen increase [12]

○ Radiological features (including serosal involvement and / or effusion) [6,7]

○ Pathological features (including necrosis, mitoses, nuclear pleomorphism) [10,13]

○ Molecular features (including molecular translocations, *WWTR1-CAMTA1* vs *TFE3-WWTR1* vs others) [11,14,15]

- To describe treatment pattern and adherence to consensus paper recommendations on EHE clinical management (including diagnostic ascertainment, treatment, follow-up) [4]

- To generate hypotheses on potential risk factors: demographics, sex, medical history (allergy, previous malignancies, immune-mediated diseases and previous immunosuppressive therapies, hormonal influence).

## Methods

### Setting and participants

**Data collection.** The EURACAN EHE registry is a prospective clinical registry that will collect data from all new patients receiving a pathological and molecularly confirmed diagnosis of EHE starting from the 1st December 2023. To this aim, an EHE-focused clinical record form (CRF) was developed through the Research Electronic Data Capture (REDCap) and designed to capture the specific features of this disease.

For each patient, data collection will include:

1- A common baseline, including full details on demographic, comorbidities (special focus on immune-mediated and gynecological diseases), concomitant medications, physical examination, systemic symptoms at presentation, tumour-related pain assessment, blood tests, pathology and molecular features (on the diagnostic specimen), EHE extension at baseline (unifocal disease, loco-regional disease, systemic metastases)2 Depending on the disease extent, detail on staging modalities, primary site and size, extension of metastatic disease (organs involved, number of lesions), presence and characterization of serosal involvement and effusion, treatment (surgery, radiation therapy, medical therapies, isolated limb perfusion, other locoregional techniques)

Although the EHE registry per se does not foresee any collection of pathological samples and / or imaging, the availability for each enrolled patient of pathological samples, massive parallel sequencing data and imaging at the contributing institution will be captured by the registry, in order to facilitate patients' identification for future dedicated studies which will imply the use of these data.

Given the peculiar biological EHE behavior, in absence of any event, a mandatory 6-monthly update will be required at all contributing centers. In the mandatory 6-monthly update, information from the last visit will be recorded regarding: physical examination, time interval of development of systemic symptoms, assessment of tumor-related pain, blood tests, update on ongoing treatment / active surveillance (including modalities and timing of radiological monitoring and radiological response).

All events will be recorded at the time of the occurrence (for unifocal disease, local recurrence and metastatic progression; for locoregional and metastatic disease, systemic progression). For every event, information will be recorded on physical examination, time interval of development of systemic symptoms, assessment of tumor-related pain assessment, blood tests (hemoglobin and fibrinogen), radiological modalities detecting progression, extent of progression (by RECIST 1.1 and not) [16], serosal involvement and / or effusion, update on ongoing treatment / active surveillance.

Patient status (alive with no evidence of disease, alive with disease, dead) and last follow up will be updated at any data entry.

The registry and therefore the electronic CRF have been shaped starting from what have been agreed and reported in the EHE consensus paper from the sarcoma community of experts, published in 2021 [4]. A registry kick-off meeting was held in Milan the 15th September 2023, involving the EHE prospective clinical registry coordination team, all the contributing centers and patients' associations (EHE Rare Cancer Charity UK, EHE Italia and EHE US) to present the registry and share common clinical practice guidelines, derived from the consensus paper, to be followed for the patients with EHE included in the registry.

**Contributing centers.** The launch of the EHE prospective clinical registry was shared with all the reference institutions belonging as full members or associated partners to EURACAN sarcoma domain (63 institutions across Europe).

The project was also opened to EU institutions not belonging to EURACAN and extra-EU institutions. In this case, in order to ensure an adequate degree of expertise in the disease, the institutions interested in joining the registry were asked to provide the number of new EHE cases (molecularly confirmed) seen each year over 3 years (2020-2021-2022) and a threshold of at least 20% of EHE national incident cases was selected for participation.

As a result of this process, 21 institutions joined the registry in September 2023, 17 of which belonging to EURACAN, from 10 countries.

## Inclusion criteria

- New patients managed by the contributing centers with a pathological EHE diagnosis performed or verified by an expert sarcoma pathologist starting from 1 December 2023 onwards and to be performed within 6 months from the registration

- Molecular confirmation of the diagnosis (*WWTR1-CAMTA1* or *YAP1-TFE3*)

- Adult patients (aged $\geq$ 18 years)

## Outcomes are listed per registry objectives

- EHE population and diseases characteristics at diagnosis (M/F ratio, average age, proportion of Charlson Comorbidity index, ECOG performance status, distribution of primary site and disease extension) [4]

- EHE natural history (cumulative incidence of local recurrence and distant metastases per different EHE extension; description of progression patterns: site of metastatic spreading, number and site of new lesions, evolution of known lesions, presence and features of serosal involvement, time to local progression or distant metastases; overall and progression free survival) [4,7]

- Treatment Effectiveness (overall response rate, by RECIST1.1 and according to clinical judgment, competitive cumulative incidence of local recurrence and distant metastases, disease-free survival, overall survival)

- Identify Prognostic and predictive factors: overall and progression free survival and treatment response

- Description of treatment pattern and quality of care: percentage of patients treated according to Consensus paper recommendations.

**Sample size.** This is an observational clinical prospective registry which implies a long-term data collection lasting until all the registry objectives are met. Considering that we expected 10 centers in Italy (more than 30 cases per year), 1 centre in United Kingdom (more than 10 cases per year), 1 centre in Poland (about 5 cases per year), 1 centre in Germany, Spain, France, Norway, Denmark, Sweden, Austria and The Netherlands (about 3–5 cases on average each centre per year), assuming that the maximum number of expected cases will be recorded in each centre we envision about 100 patients with an EHE yearly. The already established collaboration and participation of EHE patient's associations involved in the project will help in promoting the registry and fostering accrual. The precision of estimates of a categorical endpoint (e.g. proportion) or a continuous endpoint, respectively in terms of width of two-

**Table 1. Precision of the estimates expressed as 95% Confidence Interval (CI) for categorical and continuous endpoint accordingly to different sample size.**

| Target sample size | Categorical endpoint | | Continuous endpoint | |
|---|---|---|---|---|
| | max 95% CI | Width of 95% CI | max 95% CI | Width of 95% CI |
| 350 | Estimate +/- 0.05 | 0.10 | Estimate +/- 0.10*SD | 0.20*SD |
| 300 | Estimate +/- 0.06 | 0.11 | Estimate +/- 0.11*SD | 0.22*SD |
| 250 | Estimate +/- 0.06 | 0.12 | Estimate +/- 0.12*SD | 0.24*SD |
| 200 | Estimate +/- 0.07 | 0.14 | Estimate +/- 0.14*SD | 0.28*SD |
| 150 | Estimate +/- 0.08 | 0.16 | Estimate +/- 0.16*SD | 0.32*SD |
| 100 | Estimate +/- 0.10 | 0.20 | Estimate +/- 0.20*SD | 0.40*SD |
| 70 | Estimate +/- 0.12 | 0.24 | Estimate +/- 0.23*SD | 0.46*SD |

To calculate the sample size for the analytical questions involving several different variables, ad hoc analysis plans will be needed.

sided 95% Confidence Interval (CI) using the Simple Asymptotic method or in form of normal distribution for means, is based on the sample size achieved in different years during the data collection. For example, a sample size of 200 patients (e.g. EHE cases in 2 years) will achieve a maximum width of 95% CI of 0.14 (i.e. estimated proportion +/- 7%). For continuous endpoints, a sample size of 200 patients will achieve a maximum width of 95% CI of 0.28*SD (i.e. estimated mean +/- 0.14*SD, where SD = Standard Deviation).

Table 1 reports the precision of the estimates for categorical and continuous endpoint under different sample size.

**Data collection and storage.** The registry is federated thus, data are stored by the health care providers contributing to the registry. At the local level, data are pseudonymised.

## Data analyses

Statistical analyses will be performed based on a study protocol. Queries will be developed, in collaboration with clinical experts, to interrogate the EURACAN EHE registry to generate the descriptive statistics and relevant information needed to plan the analyses envisaged by the study protocol. Queries may include general description of the characteristics of the patients with EHE (M/F ratio, average age, primary site distribution, disease extension, progression patterns etc.) Due to the nature of the registry, supplementary statistical analysis plans are envisioned to reply to specific research questions (e.g. identification of prognostic/predictive factors) arising over time.

Here we report a general data analysis plan based on the objectives of the registry.

**Data analyses plan.** The data analyses will include descriptive statistics and analytical analyses. Descriptive statistics will be used to reconstruct the natural history of EHE (e.g. cumulative incidence of local recurrence and distant metastases, proportion of site of metastatic spreading, number and proportion of site of new lesions, evolution of known lesions, presence and features of serosal involvement, time to local progression or distant metastases etc.) and to report indicators about quality of care (e.g. description of staging procedures, description of treatment pattern, etc.).

Multivariable Cox's proportional hazards model and Hazard ratios (HR) for all-cause or cause-specific mortality will be used to determine independent predictors of overall survival, recurrence and progression. Variables to include in the multivariable regression model will be selected based on the results of univariable analysis and using Information Criteria, such as Akaike Information Criterion. According to the specific research question the role of potential confounders and effect modifiers will be evaluated. Sensitivity analyses will be performed to estimate the potential impact of confounders. Statistical interactions between predictors and

the variable will be tested to detect potential effect modifiers and stratified analyses will be performed in case of presence of any effect modifiers.

To assess treatment effectiveness, propensity score adjustment will be applied to multivariable models to remove bias due to real world nature of the registry.

High proportion of missing data threaten the validity of the inferences/prognostic models. Thus, a maximum of 10% of missing data will be allowed and missing data will be imputed using strategy such as unconditional/conditional mean or expectation maximum. To minimize missing information, in addition to the prospective nature of the registry, summary lists of information to be collected during clinical visits and including all the pathological details required for diagnosis were shared with all centers.

**Data quality checks.** Data quality checks aim to assess whether data values are present, if they adhere to specific standard and if they are believable in terms of:

Conformance: the data should be recorded in agreement with the correct formats (e.g. range of values, date formats);

- Completeness: the data should not have missing values or data records;

- Plausibility: the data should be believable (e.g. internal consistency; temporal and atemporal comparison);

Quality checks of the data conformance are included in the CRF in the form of predefined alerts and errors running during the data input. Moreover, several additional completeness and plausibility checks (e.g. completeness of variables for each patient, plausibility of temporal intervals between different dates, etc.) are implemented in an external web tool connected to the CRF by use of API key, capable of executing all data quality checks simultaneously. The API key is not transmitted outside the centre, it is used only locally to perform data quality checks. The results of these checks are available locally for each centre and periodically they will be summarized in reports that the registry coordination team (INT) will monitor and discuss with each centre.

**Status and timeline of the registry.** The registry initiated recruiting in December 2023. The estimated completion date is December 2033 upon agreement on the achievement of all the registry objectives. In order to promptly share with all the contributing centres and relevant stakeholders the progresses done by the EHE registry, the challenges encountered and observation derived from the analyses of patients included in the registry, especially if useful for clinical practice, yearly update meeting will be scheduled.

**Ethics statement.** In Europe, the processing of personal data for scientific research can be grounded upon several conditions of lawfulness, established in Article 9(2) of the General Data Protection Regulation (GDPR). Among these, the most relevant ones are:

- the consent of data subjects (Art. 9(2)lit. a);

- the pursuing of a substantial public interest, on the basis of Union or Member State law (Art. 9(2)lit. g);

- the pursuing of a scientific research purpose on the basis of Union or Member State law (Art. 9(2)lit. j).

In addition, another possibility lies in the so-called "Compatibility Assessment".

The GDPR allows or requires Member States to implement national specifications or derogations from certain rules set out in the GDPR, therefore the legal basis of the federated registry will be based in each participating hospital on the laws and regulations of its country.

If the legal basis of the registry is patient consent, this will be informed and written.

The EURACAN registry was approved by the Ethics Committee of the Fondazione IRCCS Istituto Nazionale dei Tumori, Milan.

## Discussion

The EHE prospective registry developed in the context of the EURACAN registry will be the first, international registry fully dedicated to an ultra-rare sarcoma such as EHE, which will serve as an example for other sarcomas and ultra-rare tumours. It has been developed with the support and in close collaboration with EHE patient's associations (EHE Rare Cancer Charity UK) and it will foresee an extensive participation of sarcoma reference centers (22) mostly belonging to the EURACAN sarcoma domain across 10 EU countries and UK, with a concrete possibility in the near future of expanding this effort further across borders.

The current project recognizes several limitations. There is a risk of limited representativeness due to the hospital-based nature of the registry and to the involvement of sarcoma reference centers only. However, the representativeness of the registry will be tested by comparing registry data with population-based data in terms of relevant variables (e.g. age, stage, prognosis). Confounders could be another issue raising from the observational nature of the registry. Adequate statistics methods (e.g. marginal structural model) will be used in case of time-varying treatments/confounders and confounding by indication (selective prescribing). Directed acyclic graphs can also be useful to identify the source of bias and will be utilized in the definitions of the path between covariates. Also, the completeness of clinical follow-up could represent a problem but active research into the patients' state of life will be guaranteed.

Despite the limitations described above, we believe that the establishment of this registry will represents a crucial step forward in the comprehension of EHE. The CRF developed for the registry, while containing the minimum dataset felt adequate for all sarcomas, has been entirely shaped for EHE, and will be able to captures all the peculiarities of this disease. The quality of the data collected within the registry will be ensured by the expertise of the joining centers but also by an agreed consistent practice in EHE management among participating clinicians, derived by the consensus paper from the worldwide sarcoma community focusing on this disease, and published in 2021. Furthermore, the semi-annual updates, while not necessarily guaranteeing the same follow-up time in each center, will increase the time windows with useful information to evaluate the time elapsed between events, increasing the quality of the data. Periodical call between the registry coordination team and the participating centers will be also scheduled to provide assistance with any possible issues which will arise in due course.

The data collected within the registry will provide a unique source on information to address some of the unresolved questions on EHE natural history, prognosis, management approach according to the different clinical scenario, treatment response and outcome. This will be instrumental for clinicians to better inform patients with EHE and tailor their treatment, but it will hopefully also provide a backbone to facilitate the development of clinical studies on new potentially active drugs and pave the way for the discussion with competent authorities for approval.

## Author Contributions

**Conceptualization:** Anna Maria Frezza, Ninna Aggerholm-Pedersen, Giuseppe Badalamenti, Giacomo G. Baldi, Simone Bonfarnuzzo, Paolo G. Casali, Claudia Giani, Alessandro Gronchi, Silvia Stacchiotti, Annalisa Trama.

**Data curation:** Anna Maria Frezza, Paolo Lasalvia.

**Funding acquisition:** Hugh Leonard, Annalisa Trama.

**Methodology:** Jean-Yves Blay, Paolo Lasalvia, Annalisa Trama.

**Project administration:** Paolo Baili, Annalisa Trama.

**Resources:** Anna Maria Frezza, Paolo Baili, Andrei Ivanescu, Silvia Stacchiotti.

**Supervision:** Anna Maria Frezza, Annalisa Trama.

**Validation:** Giuseppe Bianchi, Claudia Giani, Paolo Lasalvia, Matilde Ingrosso, Laura Monteleone.

**Visualization:** Simone Bonfarnuzzo, Claudia Giani, Paolo Lasalvia, Matilde Ingrosso, Laura Monteleone.

**Writing – original draft:** Anna Maria Frezza, Paolo Lasalvia, Silvia Stacchiotti, Annalisa Trama.

**Writing – review & editing:** Anna Maria Frezza, Hugh Leonard, Ninna Aggerholm-Pedersen, Giuseppe Badalamenti, Paolo Baili, Giacomo G. Baldi, Sebastian Bauer, Serena Bazzurri, Irene Benzonelli, Alexia Bertuzzi, Jean-Yves Blay, Giuseppe Bianchi, Simone Bonfarnuzzo, Christophe Bouvier, Kyetil Boye, Javier Martin Broto, Antonella Brunello, Domenico Campanacci, Paolo G. Casali, Carlo Cicala, Elisa Crotti, Lorenzo D'Ambrosio, Angelo Paolo Dei Tos, Nils Dieckmann, Armelle Dufresne, Stephanie Elston, Virginia Ferraresi, Stefano Gabellini, Claudia Giani, Vincenzo Giannusa, Melissa Gil Sanjines, Teresa Grassani, Alessandro Gronchi, Paolo Lasalvia, Stefan Lindskog, Nadia Hindi, Matilde Ingrosso, Andrei Ivanescu, Robin Jones, Iwona Lugowska, Julia Ketzer, Anna Mariuk-Jarema, Alessandro Mazzocca, Laura Monteleone, Carlo Morosi, Andrea Napolitano, Francesca Nardozza, Elisabetta Neri, Maria Nilsson, Andri Papakonstantinou, Sandro Pasquali, Marta Sbaraglia, Federico Scolari, Joanna Szkandera, Claudia Valverde, Bruno Vincenzi, Salvatore Vizzaccaro, Federica Zuccheri, Annalisa Trama.

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
