## [Decision Letter · Decision Letter 0]

15 Feb 2024

PONE-D-23-43229The observational EURACAN prospective clinical registry dedicated to epithelioid hemangioendothelioma:  the protocol of an international and collaborative effort on an ultra-rare entityPLOS ONE

Dear Dr. Trama,

Thank you for submitting your manuscript to PLOS ONE. After careful consideration, we feel that it has merit but does not fully meet PLOS ONE’s publication criteria as it currently stands. Therefore, we invite you to submit a revised version of the manuscript that addresses the points raised during the review process.

**ACADEMIC EDITOR: **This is an important field of research and your protocol is fairly good. Add  a sentence in the protocol describing  the procedures in case of protocol deviations and  justify why a 6 months update and not one year. Please find bellow the reccomendations of reviewer 1 and  please answer the questions .

We look forward to receiving your revised manuscript.

Kind regards,

Cristina Deppermann Fortes, PhD

Academic Editor

PLOS ONE

Journal Requirements:

https://doi.org/10.1371/journal.pone.0283071

In your revision ensure you cite all your sources (including your own works), and quote or rephrase any duplicated text outside the methods section. Further consideration is dependent on these concerns being addressed.

“Anna Maria Frezza declares institutional research funding from Advenchen Laboratories, Amgen Dompé, AROG Pharmaceuticals, Ayala Pharmaceuticals, Bayer, Blueprint Medicines, Boehriger Ingelheim, Daiichi Sankyo, Deciphera, Eisai, Eli Lilly, Epizyme Inc, Foghorn Therapeutics Inc., Glaxo, Hutchison MediPharma Limited, Inhibrx, Inc., Karyopharm Pharmaceuticals, Novartis, Pfizer, PharmaMar, PTC Therapeutics, Rain Oncology, SpringWorks Therapeutics.

Alexia Bertuzzi declares support for attending meeting and/or travels from Pharmamar and Istituto Gentili; support for scientific activities from Istituto Gentili.

Antonella Brunello declares consulting fees or serving on advisory boards for Eli Lilly, Roche, GSK, Eisai, Pharmamar, Boehringer Ingelheim, Deciphera; payment or honoraria for educational events by GSK and Pharmamar; travel grants by Pharmamar, Istituto Gentili.

Giacomo G. Baldi declares consulting fees from Eli Lilly, Pharmamar, AboutEvents; honoraria from Pharmamar, Eli Lilly, Glaxo Smith Kline, Merck Sharp & Dome, Eisai, IstitutoGentili; support for attending meetings and/or travels from Novartis, Pharmamar, Eli Lilly; participation on the advisory board from Pharmamar, Eli Lilly, Glaxo Smith Kline, Merck Sharp & Dome, Eisai.

Paolo G. Casali declares institutional research funding from Advenchen Laboratories, Amgen Dompé, AROG Pharmaceuticals, Ayala Pharmaceuticals, Bayer, Blueprint Medicines, Boehriger Ingelheim, Daiichi Sankyo, Deciphera, Eisai, Eli Lilly, Epizyme Inc, Foghorn Therapeutics Inc., Glaxo, Hutchison MediPharma Limited, Inhibrx, Inc., Karyopharm Pharmaceuticals, Novartis, Pfizer, PharmaMar, PTC Therapeutics, Rain Oncology, SpringWorks Therapeutics.

Virginia Ferraresi declares consultancy fees or honoraria from PharmaMar, Boehringer Ingelheim, Gentili e Serb Pharmaceuticals.

Matilde Ingrosso reports institutional research funding from Advenchen Laboratories, Amgen Dompé, AROG Pharmaceuticals, Ayala Pharmaceuticals, Bayer, Blueprint Medicines, Boehriger Ingelheim, Daiichi Sankyo, Deciphera, Eisai, Eli Lilly, Epizyme Inc, Foghorn Therapeutics Inc., Glaxo, Hutchison MediPharma Limited, Inhibrx, Inc., Karyopharm Pharmaceuticals, Novartis, Pfizer, PharmaMar, PTC Therapeutics, Rain Oncology, SpringWorks Therapeutics.

Laura Monteleone reports institutional research funding from Advenchen Laboratories, Amgen Dompé, AROG Pharmaceuticals, Ayala Pharmaceuticals, Bayer, Blueprint Medicines, Boehriger Ingelheim, Daiichi Sankyo, Deciphera, Eisai, Eli Lilly, Epizyme Inc, Foghorn Therapeutics Inc., Glaxo, Hutchison MediPharma Limited, Inhibrx, Inc., Karyopharm Pharmaceuticals, Novartis, Pfizer, PharmaMar, PTC Therapeutics, Rain Oncology, SpringWorks Therapeutics.

Johanna Szkandera reports participation in advisory boards or invited speaker fees for PharmaMar, Bayer, Roche, Lilly, Amgen; travel expenses coverage from PharmaMar, Roche, Lilly, Amgen, Bristol Myers Squibb; research funding from PharmaMar, Roche, Eisai.

Bruno Vincenzi reports consulting fees from Eisai, Lilly, Bayer, Deciphera, PharmaMar, Blueprint, Pfizer, GSK, Accord, Abbott and research support from PharmaMar, Novartis, Lilly

Silvia Stacchiotti declares advisory board roles with Bayer, Boehringer, Daiichi, Ikena, Nec Oncology, Pharma Essentia, Regeneron, Servier; invited speaker roles for Bayer, Boehringer, Gentili, Pharmamar; institutional research funding from Advenchen Laboratories, Amgen Dompé, AROG Pharmaceuticals, Ayala Pharmaceuticals, Bayer, Blueprint Medicines, Boehriger Ingelheim, Daiichi Sankyo, Deciphera, Eisai, Eli Lilly, Epizyme Inc, Foghorn Therapeutics Inc., Glaxo, Hutchison MediPharma Limited, Inhibrx, Inc., Karyopharm Pharmaceuticals, Novartis, Pfizer, PharmaMar, PTC Therapeutics, Rain Oncology, SpringWorks Therapeutics.

Carlo Cicala, Lorenzo D’Ambrosio, Angelo Paolo Dei Tos, Alessandro Gronchi, Nadia Hindi, Robin Jones, Iwona Lugowska, Julia Ketzer, Anna Mariuk-Jarema, Andrea Napolitano, Andri Papakonstantinou, Sandro Pasquali, Marta Sbaraglia, Federico Scolari, Elisa Crotti, Irene Benzonelli, Jean-Yves Blay, Christophe Bouvier, Javier Martin Broto, Hugh Leonard, Ninna Aggerholm-Pedersen, Andri Papakonstantinou, Giuseppe Bianchi, Federica Zuccheri, Paolo Baili, Simone Bonfarnuzzo, Domenico Campanacci, Armelle Dufresne, Sebastian Bauer, Serena Bazzurri, Stefano Gabellini, Claudia Giani, Vincenzo Giannusa, Melissa Gil-Sanjines, Teresa Grassani, Paolo Lasalvia, Stefan Lindskog, Salvatore Vizzaccaro, Nils Dieckmann, Alessandro Mazzocca, Andrei Ivanescu, Giuseppe Badalamenti, Carlo Morosi, Stephanie Elston, Kyetil Boye, Francesca Nardozza, Elisabetta Neri, Maria Nilsson and Annalisa Trama declares no conflict of interest.”

We note that one or more of the authors are employed by a commercial company

Additional Editor Comments (if provided):

This is an important field of research and your protocol is fairly good.The authors should add in the protocol that any deviations from the protocol has to be ammended with justifications and reviewed by the local the ethical committee.

Reviewers' comments:

Reviewer's Responses to Questions

**Comments to the Author**

1. Does the manuscript provide a valid rationale for the proposed study, with clearly identified and justified research questions?

Reviewer #1: Yes

2. Is the protocol technically sound and planned in a manner that will lead to a meaningful outcome and allow testing the stated hypotheses?

Reviewer #1: No

3. Is the methodology feasible and described in sufficient detail to allow the work to be replicable?

Reviewer #1: Yes

4. Have the authors described where all data underlying the findings will be made available when the study is complete?

Reviewer #1: Yes

5. Is the manuscript presented in an intelligible fashion and written in standard English?

Reviewer #1: Yes

6. Review Comments to the Author

You may also provide optional suggestions and comments to authors that they might find helpful in planning their study.

Reviewer #1: The study represents an observational registry-based cohort study which includes new pathological and molecularly confirmed diagnosis of EHE, starting from the 1st December 2023, with an estimated completion date of December 2033. The study represents the first effort of this kind, being joined by 21 sarcoma reference centers across EU and UK, spanning 10 countries. It has a projected combined accrual of 100 cases/tear. The goal is to collect full details on patients and disease features, treatment and outcome, according to common clinical practice guidelines developed and shared with all the contributing centers.

This effort is highly laudable and will be the first STARTER international registry, fully dedicated to an ultra-rare sarcoma such as EHE, which will serve as an example for other sarcomas.

One point that the authors may want to clarify or add relates to the semi-annual updates – if one particular systemic regimen has shown encouraging results in few patients – is there a mechanism in place to inform the other contributors? Meaning, can this registry serve as an interactive/active platform and have a feedback mechanism for the data analysis, not only at the end of the 10 years of observation.

Second point, a mandatory 6-monthly update is requested from the participating sites. Why only 6 month and not 1 year – as most EHE have an indolent behavior, with some multifocal liver or even lung EHE, showing no changes for years.

There are many EHE registry objectives listed. However, among prognostic and predictive factors will be important to add under the molecular features: NGS findings when available, such as secondary hits: CDKN2A/B deletion. TERT promoter mutations, etc.

Minor comments: mostly rephrasing and typos

- ‘peculiar molecular and pathologic features’ – replace ‘peculiar’ with ‘distinctive’ – in the Abstract intro as well as from the 1st sentence in the Introduction

- delete word ‘extremely’ before variable clinical behavior in the Abstract intro

- fix typo: ‘Centres’ with ‘centers’ in abstract settings, 2nd page intro and discussion

Intro: use italics for gene names and update nomenclature for gene fusions ‘-‘ replace with ‘::’

7. PLOS authors have the option to publish the peer review history of their article (what does this mean?). If published, this will include your full peer review and any attached files.

Reviewer #1: No

---

## [Author Response · Author response to Decision Letter 0]

18 Jul 2024

Reviewer #1, comments to the authors: The study represents an observational registry-based cohort study which includes new pathological and molecularly confirmed diagnosis of EHE, starting from the 1st December 2023, with an estimated completion date of December 2033. The study represents the first effort of this kind, being joined by 21 sarcoma reference centers across EU and UK, spanning 10 countries. It has a projected combined accrual of 100 cases/tear. The goal is to collect full details on patients and disease features, treatment and outcome, according to common clinical practice guidelines developed and shared with all the contributing centers. This effort is highly laudable and will be the first STARTER international registry, fully dedicated to an ultra-rare sarcoma such as EHE, which will serve as an example for other sarcomas.

One point that the authors may want to clarify or add relates to the semi-annual updates – if one particular systemic regimen has shown encouraging results in few patients – is there a mechanism in place to inform the other contributors? Meaning, can this registry serve as an interactive/active platform and have a feedback mechanism for the data analysis, not only at the end of the 10 years of observation.

Thanks to the reviewer for pointing this out. Actually, the 10-year span was meant only to provide a tentative length of the registry. Within the registry, yearly meeting will be scheduled with all the contributing centers and relevant stakeholders in order to discuss together the progresses and challenges of the registry, and to share observations, especially if useful for clinical practice, which may arise from analyses performed. It is envisaged, in fact, to carry out the analyses on a yearly basis and in a scalable way based on the number of cases registered and the years of follow-up accrued (i.e. starting from descriptive analyses to move on to predictive and prognostic analyses as the years of follow-up and the number of patients increases). The following sentence has been added at page 14, paragraph “Status and timeline of the registry” in order to improve clarity: “In order to promptly share with all the contributing centres and relevant stakeholders the progresses done by the EHE registry, the challenges encountered and observation derived from the analyses of patients included in the registry, especially if useful for clinical practice, yearly update meeting will be scheduled”. 

Second point, a mandatory 6-monthly update is requested from the participating sites. Why only 6 month and not 1 year – as most EHE have an indolent behavior, with some multifocal liver or even lung EHE, showing no changes for years.

The optimal length of the mandatory data update (which is requested in absence of events) was extensively discussed also with all the sarcoma reference centers joining this project. A 6-month interval was selected essentially for 3 reasons:

1- if it is true that, within EHE, some cases are indolent and characterized by prolonged, spontaneous disease stability, some others can slowly progress overtime or even show a very aggressive behavior. Describing the evolution of the disease in progressive cases and how this might change with treatment start is one of the goals of the registry, and we felt that this might be better and more reliably done with a 6-month interval,

2- the compulsory follow-up it is not only meant to provide data on disease status and survival, but also to monitor changes overtime in clinical features (including signs, symptoms, fertility, concomitant medications) which might influence EHE evolution and play a role as prognostic factors,

3- it was felt that a 6-monthly follow up could represent a reasonable interval in order to minimize the impact of different follow-up schedules across contributing centers in reported progression-free survival. 

There are many EHE registry objectives listed. However, among prognostic and predictive factors will be important to add under the molecular features: NGS findings when available, such as secondary hits: CDKN2A/B deletion. TERT promoter mutations, etc.

We totally agree with the reviewer on the importance of capturing these data. The registry, for its own nature, does not foresee the collection of detailed molecular data outside standard practice, nor the storage of pathological samples and / or imaging. Still, the registry captures the availability, at the contributing institution, of pathological samples (baseline and recurrences), NGS data and imaging, in order to facilitate patients’ identification future dedicated studies which will imply the use of these data. The following sentence has been added at page 8, paragraph “Data collection”: “Although the EHE registry per se does not foresee any collection of pathological samples and / or imaging, the availability for each enrolled patient of pathological samples, massive parallel sequencing data and imaging at the contributing institution will be captured by the registry, in order to facilitate patients’ identification for future dedicated studies which will imply the use of these data”.

Minor comments: mostly rephrasing and typos

- ‘peculiar molecular and pathologic features’ – replace ‘peculiar’ with ‘distinctive’ – in the Abstract intro as well as from the 1st sentence in the Introduction

- delete word ‘extremely’ before variable clinical behavior in the Abstract intro

- fix typo: ‘Centres’ with ‘centers’ in abstract settings, 2nd page intro and discussion

Intro: use italics for gene names and update nomenclature for gene fusions ‘-‘ replace with ‘::’

Thanks for spotting, done as requested.

---

## [Editor Report · Decision Letter 1]

23 Jul 2024

The observational EURACAN prospective clinical registry dedicated to epithelioid hemangioendothelioma:  the protocol of an international and collaborative effort on an ultra-rare entity

PONE-D-23-43229R1

Dear Dr. Annalisa Trama,

We’re pleased to inform you that your manuscript has been judged scientifically suitable for publication and will be formally accepted for publication once it meets all outstanding technical requirements.

Kind regards,

Cristina Deppermann Fortes, PhD

Academic Editor

PLOS ONE

Additional Editor Comments (optional):

Please add clearly the objectives of the Registry and state its limitations as suggested by the reviewer. 

Reviewers' comments:

The study represents an observational registry-based cohort study which includes new pathological and molecularly confirmed diagnosis of EHE, starting from the 1st December 2023, with an estimated completion date of December 2033. The study represents the first effort of this kind, being joined by 21 sarcoma reference centers across EU and UK, spanning 10 countries. It has a projected combined accrual of 100 cases/tear. The goal is to collect full details on patients and disease features, treatment and outcome, according to common clinical practice guidelines developed and shared with all the contributing centers.

This effort is highly laudable and will be the first STARTER international registry, fully dedicated to an ultra-rare sarcoma such as EHE, which will serve as an example for other sarcomas.

One point that the authors may want to clarify or add relates to the semi-annual updates – if one particular systemic regimen has shown encouraging results in few patients – is there a mechanism in place to inform the other contributors? Meaning, can this registry serve as an interactive/active platform and have a feedback mechanism for the data analysis, not only at the end of the 10 years of observation.

Second point, a mandatory 6-monthly update is requested from the participating sites. Why only 6 month and not 1 year – as most EHE have an indolent behavior, with some multifocal liver or even lung EHE, showing no changes for years.

There are many EHE registry objectives listed. However, among prognostic and predictive factors will be important to add under the molecular features: NGS findings when available, such as secondary hits: CDKN2A/B deletion. TERT promoter mutations, etc.

Minor comments: mostly rephrasing and typos

- ‘peculiar molecular and pathologic features’ – replace ‘peculiar’ with ‘distinctive’ – in the Abstract intro as well as from the 1st sentence in the Introduction

- delete word ‘extremely’ before variable clinical behavior in the Abstract intro

- fix typo: ‘Centres’ with ‘centers’ in abstract settings, 2nd page intro and discussion

Intro: use italics for gene names and update nomenclature for gene fusions ‘-‘ replace with ‘::’

---

## [Editor Report · Acceptance letter]

5 Aug 2024

PONE-D-23-43229R1 

PLOS ONE

Dear Dr. Trama, 

I'm pleased to inform you that your manuscript has been deemed suitable for publication in PLOS ONE. Congratulations! Your manuscript is now being handed over to our production team.

Kind regards, 

on behalf of

Dr. Cristina Deppermann Fortes 

Academic Editor

PLOS ONE